# Prediction of Parkinson’s Disease Risk Based on Genetic Profile and Established Risk Factors

**DOI:** 10.3390/genes12081278

**Published:** 2021-08-20

**Authors:** Paraskevi P. Chairta, Andreas Hadjisavvas, Andrea N. Georgiou, Maria A. Loizidou, Kristia Yiangou, Christiana A. Demetriou, Yiolanda P. Christou, Marios Pantziaris, Kyriaki Michailidou, Eleni Zamba-Papanicolaou

**Affiliations:** 1Neurology Clinics, The Cyprus Institute of Neurology and Genetics, Nicosia 2371, Cyprus; yiolandac@cing.ac.cy (Y.P.C.); pantzari@cing.ac.cy (M.P.); ezamba@cing.ac.cy (E.Z.-P.); 2The Cyprus School of Molecular Medicine, The Cyprus Institute of Neurology and Genetics, Nicosia 2371, Cyprus; ahsavvas@cing.ac.cy (A.H.); loizidou@cing.ac.cy (M.A.L.); kristiay@cing.ac.cy (K.Y.); kyriakimi@cing.ac.cy (K.M.); 3Department of Cancer Genetics, Therapeutics & Ultrastructural Pathology, The Cyprus Institute of Neurology and Genetics, Nicosia 2371, Cyprus; 4Department of Hygiene and Epidemiology, School of Medicine, University of Ioannina, 45110 Ioannina, Greece; andrea.gewrgiou@gmail.com; 5Department of Primary Care and Population Health, University of Nicosia Medical School, Nicosia 2408, Cyprus; demetriou.chri@unic.ac.cy; 6Biostatistics Unit, The Cyprus Institute of Neurology and Genetics, Nicosia 2371, Cyprus

**Keywords:** Parkinson’s disease, polygenic risk score, PRS, predictive model, genetic variants, SNPs, environmental factors, case-control study, Cypriot population

## Abstract

Background: Parkinson’s disease (PD) is a neurodegenerative disorder, and literature suggests that genetics and lifestyle/environmental factors may play a key role in the triggering of the disease. This study aimed to evaluate the predictive performance of a 12-Single Nucleotide Polymorphisms (SNPs) polygenic risk score (PRS) in combination with already established PD-environmental/lifestyle factors. Methods: Genotypic and lifestyle/environmental data on 235 PD-patients and 464 controls were obtained from a previous study carried out in the Cypriot population. A PRS was calculated for each individual. Univariate logistic-regression analysis was used to assess the association of PRS and each risk factor with PD-status. Stepwise-regression analysis was used to select the best predictive model for PD combining genetic and lifestyle/environmental factors. Results: The 12-SNPs PRS was significantly increased in PD-cases compared to controls. Furthermore, univariate analyses showed that age, head injury, family history, depression, and Body Mass Index (BMI) were significantly associated with PD-status. Stepwise-regression suggested that a model which includes PRS and seven other independent lifestyle/environmental factors is the most predictive of PD in our population. Conclusions: These results suggest an association between both genetic and environmental factors and PD, and highlight the potential for the use of PRS in combination with the classical risk factors for risk prediction of PD.

## 1. Introduction

Parkinson’s disease (PD) is a progressive neurodegenerative movement disorder and the second most common after Alzheimer’s disease, worldwide [1]. Selective loss or death of dopamine secreting neurons of the substantia nigra, and Lewy bodies accumulation in the spinal cord and brain are key pathological feature of the disease [2]. PD is characterised by motor symptoms such as rigidity, bradykinesia, and resting tremor as well as non-motor symptoms [2]. The prevalence of PD varies by age as it affects ~0.3% of the general population, ~1% of the population over of 60 years, and 3.5% of the population over 85 years [1,3]. Up to date, the etiology of PD is still unclear [2]. However, PD is considered as a multifactorial disorder and previous epidemiological studies suggest that both genetic and environmental factors play an essential role in the triggering of the disease [2,3,4].

Over the years, more than 90 genetic variants that are associated with sporadic PD, progression and age at onset have been identified through multiple Genome Wide Association Studies (GWAS) [5]. Despite the large number of the genetic studies and the reported variants in PD, some of the variants have a small effect on disease risk and an important proportion of the overall genetic contribution to PD risk is not clearly understood [1]. In addition, previous studies reported that environmental, including lifestyle, factors may also play a role in the development of the disease. Environmental factors, such as diet (e.g., dairy products, soft drinks, and red meat consumption), depression, exposure to pesticides, rural living and head injury were positively associated with PD, whereas smoking, alcohol, coffee consumption, and physical activity were inversely associated with the disease [3,6,7,8].

Although the already published genetic and epidemiological studies explain a substantial part of genetic as well as phenotypic variability and etiology of PD, a large fraction of genetic and environmental contribution remains to be studied [1]. Genetic and environmental factors may interact with each other in a complex manner, increasing the risk for the development of PD [2,3,4]. Several studies indicated that polygenic risk scores (PRS), which combine the effect of multiple genetic variants, can capture the overall genetic background of complex traits and diseases, including PD [1,4,9,10]. Furthermore, a combination of the genetic background with environmental findings may provide a better understanding of the disease. Although, a small GWAS was previously performed in the Greek population and some of the investigated SNPs (rs6599389 and rs356220) had the same OR direction with the results of our study in the Greek-Cypriot population, PRS was not calculated [11]. Therefore, the aim of this pilot study was to evaluate the predictive performance of a PRS consisting of 12-SNPs, which have been previously associated with PD in GWAS, and to test the combined effect of the PRS and already established risk factors on PD risk. This is a proof-of-concept study that aims to explore our population. Despite that the sample size of our study in this population is relatively small; this study provides important results and will initiate the investigation of combined environmental and genetic factors of PD in the Greek-Cypriot population. Future studies with increased sample size and number of SNPs could redefine future models and will have the potential to be evaluated for their clinical utility.

## 2. Materials and Methods

### 2.1. Dataset

This investigation used genotypic and demographic/lifestyle data obtained from a previous case-control study carried out in the Greek-Cypriot population (235 unrelated PD patients and 464 age and sex matched unrelated healthy controls) [3]. All patients were recruited in the study after clinical diagnosis of PD. Demographic/lifestyle data from PD cases and controls were collected through a personal interview. The study was approved by the Cyprus National Bioethics Committee (ΕΕΒΚ/ΕΠ/2014/29) and all subjects gave written informed consent in accordance with the 1964 Declaration of Helsinki. Genotypic data of 12 (rs12185268, rs10513789, rs6599389, rs356220, rs7617877, rs17115100, rs13312, rs1801582, rs4837628, rs823118, rs356182, rs17649553) out of 13 SNPs that were genotyped by Georgiou et al. [3] were used for the current study. These SNPs were reported to be associated with PD in previous GWAS or interaction studies [12,13,14,15,16]. One SNP was excluded from our study due to discrepancy between the genotypes. Detailed information on the study’s methodology and original SNP selection can be found in the original publication [3]. Based on the previous study, the selected SNPs have been associated with PD (*p* ≤ 5 × 10^−8^) in at least one out of the five large GWAS meta-analysis studies for PD in the European population and had 0.81 > OR > 1.23 and MAF > 5%.

### 2.2. PRS Calculation

A weighted PRS based on 12 SNPs previously related with PD [3] (Table 1) was calculated for each individual. The PRS was calculated following the approach previously described in Mavaddat et al. [17] and using the formula PRS = β_1_x_1_ + β_2_x_2_ +...+ β_k_x_k_ +…+ β_n_x_n_; where β is the log(OR) of each SNP from previous published studies and x is the i^th^ SNP dosage (0,1,2) of each individual in our dataset.

### 2.3. Selected Demographic Data

Fifteen demographic variables were selected and assessed for this study: age (years), gender (female/male), outdoor work (yes/no), pesticides or toxic substances (yes/no), pesticides (yes/no), well water drinking (yes/no), head injury (yes/no), family history of PD (yes/no), hypertension (yes/no), statin use (yes/no), depression (yes/no), smoking-current or ever (yes/no), physical activity (yes/no), Body Mass Index (BMI) (kg/m^2^), and coffee consumption (yes/no).

### 2.4. Imputation

Complete observation of the missing data was carried out using imputation packages in R. Briefly, genomic prediction was carried out using Ridge Regression Best Linear Unbiased Predictor (rrBLUP) implemented in R package rrBLUP [18], while demographic data imputation was performed by Multivariable Imputation via Chained Equations (MICE) package [19].

### 2.5. Statistical Analysis

A series of packages which work under R software (version 3.6.3) were used. Univariate logistic regression analysis was applied in order to assess the association of each previously established PD risk factor and PD status. Area Under Curve (AUC) was also calculated in order to measure the ability of a risk factor to distinguish between PD patients and controls. In addition, logistic regression analysis was also applied in order to assess the relationship between the PRS and PD status. A *p*-value of < 0.05 was considered as statistically significant. Odds ratio (OR) and 95% confidence intervals (CI) were calculated.

Continuous variables, PRS and BMI were stratified into quartiles and ORs of each quartile were assessed using logistic regression with the 25–50% (second quartile) and 20–24.94 (normal weight) ranges as references, respectively. PRS was standarised based on the control values.

Stepwise-regression analysis was used to select the best performing variables in the predictive models, using all the variables from the univariate logistic regression analyses. Stepwise regression was performed using the statistical software R following three approaches; forward selection, backward elimination, and bidirectional elimination. AUC and goodness-of-fit (GOF) were calculated for all the different models. PRS was adjusted by all covariates and possible cofounders.

Cases and controls were stratified into deciles based on the significant variables/risk factors, obtained from the predicted probabilities of the multivariate model. The ORs of extreme deciles were evaluated using logistic regression with a reference range of 40–50%. Plots were designed using ggplot2 function in R.

## 3. Results

### 3.1. Study Participants

This study included the genotypic and demographic data of 235 PD patients with a mean ± standard deviation (SD) age of 66.5 ± 10.5 years and 464 age and sex matched controls with a mean ± SD age of 65 ± 10.7.

### 3.2. Parkinson’s Disease Risk

In this study, we assessed the predictive performance of a PRS consisting of 12-SNPs and tested the combined effect of the PRS and already established PD risk factors. All demographic data before and after the imputation are shown in Table 2 and Appendix A, respectively. Univariate logistic regression analyses showed that four risk factors (age (*p* = 1.25 × 10^−5^; OR: 1.04; 95% CI: 1.02–1.05), head injury (*p* = 3.21 × 10^−3^; OR: 1.67; 95% CI: 1.19–2.36), family history (*p* = 4.55 × 10^−14^; OR: 5.27; 95% CI: 3.44–8.19) and depression (*p* ≤ 2.00 × 10^−16^; OR: 7.47; 95% CI: 4.98–11.37)) were positively associated with PD risk, while BMI at enrollment time was inversely associated with PD risk (*p* = 4.03 × 10^−6^; OR: 0.91; 95% CI: (0.87–0.94)) (Table 3 and Appendix A). Logistic regression analysis evidenced that the 12 SNPs-PRS was significantly associated with PD (*p* = 1.87 × 10^−2^; OR: 1.39; 95% CI: (1.06–1.84)) (Table 3, Figure 1, Appendix A).

The percentage of cases and controls in PRS score and BMI quartiles and their ORs were also assessed. Table 4 and Table 5 show the results before the imputation while Appendix A and Appendix A show the results after the imputation. PRS quartiles analysis showed that participants with the lowest quartile exhibited a non-significant lower risk of PD (OR: 0.74; 95% CI: 0.46−1.19; *p* = 2.21 × 10^−1^) compared to the reference quartile. On the contrary, participants in the highest quartile exhibited a non-significant increased risk of PD (OR: 1.14; 95% CI: 0.74–1.76; *p* = 5.44 × 10^−1^). BMI quartiles analysis showed that obese (24.95–29.94) participants have approximately 2.5 times less risk to develop PD compared to normal BMI participants (OR: 0.42; 95% CI: 0.25–0.69; *p*-value: 7.64 × 10^−4^), whereas underweight (<20) participants are 3 times more likely to develop PD compared again to normal (OR: 2.99; 95% CI: 1.21–7.88; *p*-value: 2.07 × 10^−2^). These trends are similar in the imputed data as well.

Stepwise regression model analysis was carried out in forward selection, backward elimination, and bidirectional elimination approaches. All approaches resulted in the same model and suggested that the best predictive model for PD in our study includes eight independent variables; PRS score, age, gender, head injury, family history, depression, smoking, and BMI (Table 6 and Appendix A, before and after imputation, respectively).

All analyses of imputed and non-imputed data yielded similar results.

Distribution of cases and controls in deciles of the final multivariate model was investigated (Table 7 and Figure 2) and OR by decile was also calculated (Figure 3) on the non-imputed data. The OR of the first decile was 0.14 (95% CI: 0.03–0.47) with a *p*-value of 3.66 × 10^−3^. This decile includes 1.9% of cases versus 16.2% of controls. This trend was also observed in the second decile with 14.2% controls and 2.6% cases (OR: 0.21; 95% CI: 0.06–0.66; *p*-value: 1.14 × 10^−2^). Interestingly, in the last two deciles (9th and 10th) the ORs were 4.98 (95%CI: 2.20–11.84) and 12 (95% CI: 4.76–33.08), respectively. In addition, the percentages of cases and controls were inversed, with the proportion of cases to be higher than the proportion of controls. These analyses were also carried out on the imputed data and results are shown in Appendix A. These results show that the multivariate model enables the stratification of the population according to the risk of developing PD.

## 4. Discussion

A large number of studies suggest that a combination of genetic and environmental/lifestyle factors play a key role in the triggering of PD [20]. Although, several GWAS studies were carried out on PD, 90 genetic variants, which have been reported to be associated with the disease in the latest GWAS, explain only approximately 16% of the PD burden [4]. In addition, epidemiological studies reported various environmental/lifestyle factors that are either positively or negatively associated with the development of the disease [4]. The incorporation of these genetic and non-genetic factors in predictive models may help in the identification of individuals with a higher risk to develop PD. The aim of this study was to investigate the predictive performance of a PRS consisting of 12 SNPs that have been previously associated with PD, and also to assess the association between already established environmental PD risk factors and PD risk. Despite that the largest PD GWAS meta-analysis has reported 90 risk loci associated with PD risk in the European population [16], in our study we assess only 12 SNP that were previously selected and investigated in our population. We carried out an evaluation/replication study because the number of Greek-Cypriot patients with PD was relatively small and thus statistical power for a discovery study could not be reached.

Through this study, we assessed some risk factors that were previously reported to be associated with the development of PD and we found that age, head injury, family history, and depression were positively associated with PD, while BMI was inversely associated with the disease (Table 3 and Appendix A). Interestingly, these results are consistent with the results of previous studies (head injury (OR:1.55; 95% CI:1.33–1.81)) [21]; family history (RR:4.45; 95% CI:3.39–5.83) [22]; depression (OR:15.1; 95% CI:5.64–40.78) [8]). In addition, Chen et al. [23] in a meta-analysis study suggested that being overweight may decrease the risk of developing PD. In our analysis for BMI, it was observed that being obese was significantly associated with decreased risk for PD, while being underweight was significantly associated with increased risk for PD, compared with individuals with normal BMI. These results suggest that BMI might be associated with the development of PD. Our findings are in concordance with Noyce et al. [24], but of course these associations might also be a result of PD side effects, e.g., due to poor nutrition.

Furthermore, a PRS consisting of 12-SNPs that were genotyped in a previous study by Georgiou et al. [3] was also calculated. Logistic regression analysis showed that the 12-SNPs PRS was significantly associated with PD status (OR:1.39; 95% CI:1.06–1.84). In addition, division of PRS in quartiles highlighted that individuals with a higher PRS score have a higher risk to develop PD while individuals with lower PRS score have a lower risk. A recent study by Jacobs et al. demonstrated that individuals which are in the highest PRS decile had about 3.5 times higher risk to develop PD compared to the individuals in the lowest PRS decile [4]. In a previous study, Escott-Price et al. [9] reported that PRS is correlated with age of onset in PD as the average of PRS was significantly higher in patients with early onset compared to late onset [9]. In another study, Ibanez et al. replicated the results using GWAS loci from Nalls et al. [16] and suggested that the genetic plays an essential role both in PD risk and its age of onset [1]. On the other hand, Butcher et al. used common variants that are associated with PD and showed that the PRS of those variants is not significantly associated with PD risk [25]. In a more recent study, Nalls et al. performed 2-stages PRS analysis using ~90 and ~2000 variants from NeuroX-dbGaP dataset and showed that AUC of PRS with the larger number of variants was better and based on their calculations, these PRSs explain ~16% and 26% of PD heritability [26]. Similar to our results, individuals with PRS values in the highest quartile had higher risk to develop PD while individuals with PRS values in the lowest quartile had lower risk compared to the reference range [26]. Furthermore, Paul et al. used 23 GWAS SNPs and suggested an association between the PRS and faster cognitive dysfunction and progression of motor symptoms [27]. In addition, Iwaki et al. [28] showed that PRS may modify the penetrance and age of onset in *LRRK2* p.G2019S carriers. Up to date, the largest Genome-Wide Polygenic Risk Score (GPRS) was carried out by Han et al. [29] using data from ~80,000 individuals and 6.2 million variants and showed that the GPRS is associated with age of onset, PD risk, and UPDRS scores.

In this study, we performed stepwise-regression analysis using the three different approaches in order to estimate the best predictive model for PD in our population. All three approaches suggested that the best predictive model includes PRS as well as seven additional independent factors (age, gender, head injury, family history, depression, smoking, and BMI). A similarly designed study carried out by Jacobs et al. [4] using data from the UK biobank demonstrated that family history, not-smoking, low-alcohol consumption, sleepiness, depression, family history with dementia, early menarche and epilepsy are strongly associated with PD. No essential differences were observed when all the significant risk factors were combined. Interestingly, model performance was moderately improved with the inclusion of the PRS in the PREDICT-PD algorithm [4].

We also investigated the distribution of cases and controls using the final multivariate model. In the lowest deciles, the percentage of controls was large, and the percentage of cases was small. On the other hand, in the highest deciles the percentages of cases and controls were inversed, with the proportion of cases to be greater than the proportion of controls. These results highlight the capability of this multivariate model to stratify the population according to the risk of developing PD.

The main limitations of our study are the small number of cohort and the small number of SNPs that were used for the PRS calculation, which lead to lower power of the study and minimal ability of genetic factors to differentiate case from controls (e.g., AUC of PRS; 0.55). Selection bias might be also included as a small single population was sampled and some members of the population are more likely to be included than others. In addition, as several data were collected through Yes/No answers from the participants, interview bias might be also included. However, this study is important as it is the first pilot study which aims to evaluate the predictive performance of a PRS and design a predictive model for PD in the Greek-Cypriot population, a relatively small population. After the appropriate evaluations, a future model with more SNPs and individuals could be used as a complementary diagnostic method. In addition, this study initiated the investigation of combined genetic and environmental factors of PD in our population.

## 5. Conclusions

In conclusion, these results suggest an association between five environmental/lifestyle factors as well as a 12-SNPs PRS and risk of PD. In addition, the model, which combines eight independent factors, could be useful for the calculation of a risk score predictive of PD in the Greek-Cypriot population.

Therefore, this may facilitate a better understanding of gene-gene as well as gene-environment interactions in the development of PD. Further investigation with a larger cohort and a PRS with additional variants may increase the statistical power and confirm that the combination of these factors could potentially be used for predictive testing.

## Figures and Tables

**Figure 1 genes-12-01278-f001:**
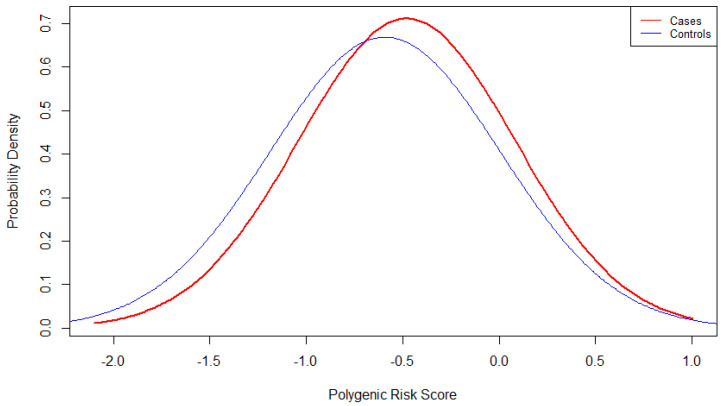
PRS (polygenic risk score) distribution between PD (Parkinson’s disease) cases and controls before imputation. This plot shows the probability density versus PRS in cases and controls.

**Figure 2 genes-12-01278-f002:**
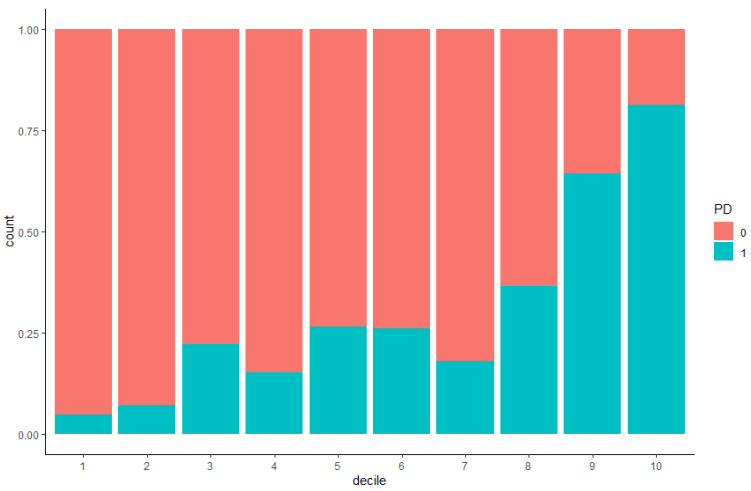
Cases and controls distribution in deciles using the multivariate model before imputation. The distribution of cases and controls are described in blue and orange, respectively.

**Figure 3 genes-12-01278-f003:**
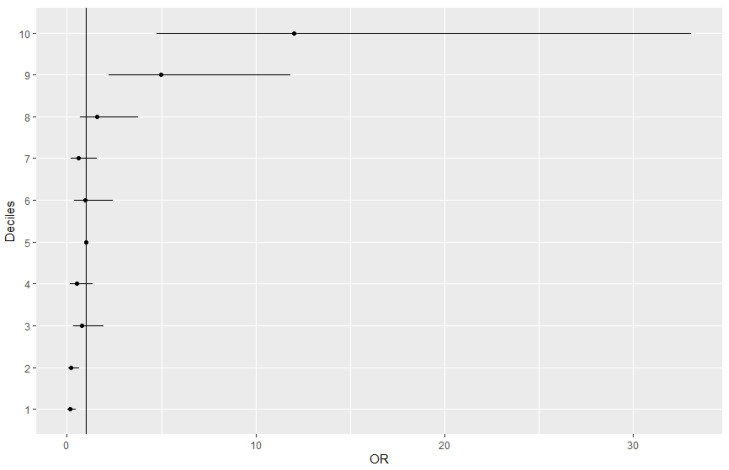
OR by decile of the multivariate model before imputation.

**Table 1 genes-12-01278-t001:** Odds ratio (OR) and 95% confidence intervals (95% CI) for the associations between 12 Single Nucleotide Polymorphisms (SNPs) and Parkinson’s disease (PD) risk.

				Georgiou et al. [3]				Previous Studies				
#	SNP	Nearest Gene	Alleles	Minor Allele	MAF	OR (95% CI) *	*p*-Value	European MAF *	Minor Allele	OR (95% CI) **	*p*-Value	References
1	rs12185268 ^&^	*SPPL2C*	A/G	G	0.26	0.69 (0.52–0.90)	0.006	0.22	G	0.77 (0.72–0.82)	2.72 × 10^−14^	[12]
2	rs10513789	*MCCC1*	T/G	G	0.18	1.09 (0.82–1.45)	0.57	0.20	G	0.80 (0.75–0.86)	2.67 × 10^−10^	[12]
3	rs6599389 ^&^	*TMEM175*	G/A	A	0.09	1.50 (1.04–2.16)	0.03	0.07	A	1.31 (1.19–1.44)	3.87 × 10^−8^	[12]
4	rs356220 ^&^	*SNCA*	C/T	T	0.37	1.33 (1.05–1.67)	0.02	0.37	T	1.29 (1.22–1.36)	2.29 × 10^−19^	[12]
5	rs7617877 ^&^	*LINC00693*	A/G	A	0.26	1.03 (0.80–1.34)	0.8	0.35	A	1.23 (1.13–1.33)	4.49 × 10^−7^	[12,13]
6	rs17115100	*CYP17A1*	G/T	T	0.09	1.06 (0.74–1.53)	0.75	0.1	T	0.80 (NR)	7.44 × 10^−8^	[12,14]
7	rs13312	*USP24*	C/G	G	0.14	1.68 (1.23–2.28)	0.001	0.24	G	0.76 (0.66–0.86)	Not reported	[12]
8	rs1801582	*PARK2*	C/G	G	0.16	1.08 (0.80–1.46)	0.63	0.16	G	0.79 (0.64–0.97)	Not reported	[12]
9	rs4837628 ^&^	*BRINP1*	C/T	C	0.34	0.89 (0.69–1.14)	0.36	0.42	C	0.79 (0.72–0.87)	1.07 × 10^−6^	[12,15]
10	rs823118 ^&^	*NUCKS1*	C/T	C	0.37	0.79 (0.62–1.01)	0.056	0.44	T ***	1.12 (NR)	1.66 × 10^−^^16^	[16]
11	rs356182 ^&^	*SNCA*	A/G	G	0.33	1.24 (0.98–1.57)	0.076	0.36	A ***	0.76 (NR)	4.16 × 10^−73^	[16]
12	rs17649553 ^&^	*MAPT*	C/T	T	0.26	0.71 (0.54–0.93)	0.013	0.21	T ***	0.77 (0.74–0.80)	2.37 × 10^−48^	[16]

**^&^** SNPs with the same direction OR, * MAF of European population submitted in 1000 Genome Project (dbSNP), ** OR for minor allele, *** Effect allele, MAF; Minor Allele Frequency, NR: Not Reported.

**Table 2 genes-12-01278-t002:** Demographic characteristics and lifestyle/environmental risk factors of Cypriot PD cases and controls before imputation.

			Controls	Cases
	**Age (Years, Mean ± SD)**		65 ± 10.7	66.5 ± 10.5
	**Age of Onset (Mean ± SD)**			60.4 ± 11.4
	**BMI**		28.28 ± 5.01	26.29 ± 4.15
	**Variable**	**Total *n* (%)**	**Controls *n* (%)**	**Cases *n* (%)**
**Gender**	Male	359 (51.36)	231 (49.78)	128 (54.47)
	Female	340 (48.64)	233 (50.22)	107 (45.53)
	NA	0 (0)	0 (0)	0 (0)
**Outdoor Work**	No	512 (73.25)	348 (75)	164 (69.79)
	Yes	168 (24.03)	103 (22.2)	65 (27.66)
	NA	19 (2.72)	13 (2.80)	6 (2.55)
**Pesticides or Toxic Substances**	No	384 (54.94)	261 (56.25)	123 (52.34)
	Yes	315 (45.06)	203 (43.75)	112 (47.66)
	NA	0 (0)	0 (0)	0 (0)
**Pesticides**	No	462 (66.09)	309 (66.59)	153 (65.11)
	Yes	237 (33.91)	155 (33.41)	82 (34.89)
	NA	0 (0)	0 (0)	0 (0)
**Well Water Drinking**	No	296 (42.35)	201 (43.32)	95 (40.43)
	Yes	389 (55.65)	258 (55.6)	131 (55.74)
	NA	14 (2.00)	5 (1.08)	9 (3.83)
**Head Injury**	No	488 (69.81)	344 (74.14)	144 (61.28)
	Yes	199 (28.47)	117 (25.22)	82 (34.89)
	NA	12 (1.72)	3 (0.65)	9 (3.83)
**Family history of PD**	No	582 (83.26)	425 (91.59)	157 (66.81)
	Yes	112 (16.02)	38 (8.19)	74 (31.49)
	NA	5 (0.72)	1 (0.22)	4 (1.70)
**Hypertension**	No	404 (57.8)	268 (57.76)	136 (57.87)
	Yes	284 (40.63)	191 (41.16)	93 (39.57)
	NA	11 (1.57)	5 (1.08)	6 (2.55)
**Statin Use**	No	463 (66.24)	313 (67.46)	150 (63.83)
	Yes	222 (31.76)	143 (30.82)	79 (33.62)
	NA	14 (2.00)	8 (1.72)	6 (2.55)
**Depression**	No	504 (72.1)	393 (84.7)	111 (47.23)
	Yes	140 (20.03)	45 (9.7)	95 (40.43)
	NA	55 (7.87)	26 (5.60)	29 (12.34)
**Smoking (Current or ever)**	No	350 (50.07)	222 (47.84)	128 (54.47)
	Yes	332 (47.5)	233 (50.22)	99 (42.13)
	NA	17 (2.43)	9 (1.95)	8 (3.40)
**Physical Activity**	No	481 (68.81)	331 (71.34)	150 (63.83)
	Yes	180 (25.75)	128 (27.59)	52 (22.13)
	NA	38 (5.44)	5 (1.08)	33 (14.04)
**Coffee Consumption**	No	32 (4.58)	19 (4.09)	13 (5.53)
	Yes	654 (93.56)	439 (94.61)	215 (91.49)
	NA	13 (1.86)	6 (1.29)	7 (2.98)

**Table 3 genes-12-01278-t003:** Univariate-logistic regression analysis of lifestyle/environmental exposure risk factors before imputation.

	*p*-Value	OR (95% CI)	AUC
PRS	1.87 × 10^−2^ **	1.39 (1.06–1.84)	0.55
Age	1.25 × 10^−5^ **	1.04 (1.02–1.05)	0.62
Gender (female)	2.42 × 10^−1^	0.83 (0.6–1.13)	0.52
Outdoor work	1.14 × 10^−1^	1.34 (0.93–1.92)	0.53
Pesticides or toxic substances	3.27 × 10^−1^	1.17 (0.85–1.6)	0.52
Pesticides	6.95 × 10^−1^	1.07 (0.77–1.48)	0.51
Well water drinking	6.63 × 10^−1^	1.07 (0.78–1.48)	0.51
Head injury	3.21 × 10^−3^ **	1.67 (1.19–2.36)	0.55
Family history of PD	4.55 × 10^−14^ **	5.27 (3.44–8.19)	0.62
Hypertension	8.02 × 10^−1^	0.96 (0.69–1.32)	0.51
Statin use	4.08 × 10^−1^	1.15 (0.82–1.61)	0.52
Depression	<2.00 × 10^−16^ **	7.47 (4.98–11.37)	0.68
Smoking (current or ever)	6.18 × 10^−2^	0.74 (0.53–1.01)	0.54
Physical activity	5.68 × 10^−1^	0.9 (0.61–1.3)	0.51
BMI	4.03 × 10^−6^ **	0.91 (0.87–0.94)	0.62
Coffee consumption	3.65 × 10^−1^	0.72 (0.35–1.51)	0.51

****** *p* value < 0.05.

**Table 4 genes-12-01278-t004:** ORs, 95% CI and distribution of cases and controls in PRS quartiles before imputation.

**PRS Score Quartiles**	**Controls *n* (%)**	**Cases *n* (%)**	***p*-Value**	**OR (95% CI)**
0–25	110 (24.2)	42 (18.3)	2.21 × 10^−1^	0.74 (0.46–1.19)
* 25–50	117 (25.7)	60 (26.2)		
50–75	112 (24.6)	59 (25.8)	9.05 × 10^−1^	1.03 (0.66–1.60)
75–100	116 (25.5)	68 (29.7)	5.44 × 10^−1^	1.14 (0.74–1.76)
<NA>	9 (1.9)	6 (2.6)		

* Reference range.

**Table 5 genes-12-01278-t005:** ORs, 95% CI, and distribution of cases and controls in BMI quartiles before imputation.

**BMI Categories**	**Controls *n* (%)**	**Cases *n* (%)**	***p*-Value**	**OR (95% CI)**
Underweight (<20)	8 (1.7)	14 (6)	2.07 × 10^−2^ **	2.99 (1.21–7.88)
* Normal (20–24.94)	99 (21.3)	58 (24.7)		
Obesity (24.95–29.94)	133 (28.7)	33 (14)	7.64 × 10^−4^ **	0.42 (0.25–0.69)
Overweight (>29.95)	174 (37.5)	83 (35.3)	3.33 × 10^−1^	0.81 (0.54–1.24)
<NA>	50 (10.8)	47 (20)		

* Reference range; ** *p* value < 0.05.

**Table 6 genes-12-01278-t006:** Stepwise-regression analysis by forward selection, backward elimination, and bidirectional elimination approaches using data before imputation.

	*p*-Value	OR (95% CI)
(Intercept)	3.49 × 10^−1^	0.36 (0.04–3.06)
PRS score	6.19 × 10^−2^	1.45 (0.99–2.15)
Current age	1.62 × 10^−3^	1.04 (1.01–1.06)
Gender	1.41 × 10^−2^	0.51 (0.3–0.87)
Head injury	3.86 × 10^−3^	1.99 (1.25–3.18)
Family history of PD	1.44 × 10^−5^	3.48 (1.99–6.14)
Depression	1.13 × 10^−13^	7.01 (4.23–11.86)
Smoking (current or ever)	1.33 × 10^−1^	0.67 (0.4–1.13)
BMI	1.16 × 10^−3^	0.92 (0.87–0.97)
AUC (95% CI)	0.79 (0.75–0.83)	
GOF	0.09	

**Table 7 genes-12-01278-t007:** ORs, 95% CI and distribution of cases and controls in deciles of the final multivariate model before imputation.

Deciles	OR (95% CI)	*p*-Value	Controls	Controls (%)	Cases	Cases (%)
0–10%	0.14(0.03–0.47)	3.66 × 10^–3^ **	59	16.2	3	1.9
10–20%	0.21(0.06–0.66)	1.14 × 10^–2^ **	52	14.2	4	2.6
20–30%	0.79(0.32–1.96)	6.11 × 10^–1^	42	11.5	12	7.8
30–40%	0.50(0.17–1.35)	1.81 × 10^–1^	39	10.7	7	4.5
40–50% *			36	9.9	13	8.4
50–60%	0.98(0.39–2.45)	9.61 × 10^–1^	34	9.3	12	7.8
60–70%	0.61(0.23–1.57)	3.10 × 10^–1^	41	11.2	9	5.8
70–80%	1.59(0.69–3.79)	2.82 × 10^–1^	33	9	19	12.3
80–90%	4.98(2.20–11.84)	1.70 × 10^–4^ **	20	5.5	36	23.4
90–100%	12.00(4.76–33.08)	4.26 × 10^–7^ **	9	2.5	39	25.3

* Reference range; ****** *p* value < 0.05.

## Data Availability

Publicly available datasets were analyzed in this study. This data can be found here: https://www.ebi.ac.uk/eva/?eva-study=PRJEB32182 (accessed on 15 October 2020). Georgiou et al. 2019 [3].

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
