# Peer review of "Prediction of Parkinson’s Disease Risk Based on Genetic Profile and Established Risk Factors"

_genes, 2021, doi:10.3390/genes12081278_

Round 1
Reviewer 1 Report
As stated in the introduction section, this paper is a proof of concept study examining whether a regression model using genetic and demographic data can be predictive for PD. It is a small study limited to a single population.
Some points to address:
- Demographic data is collected at interview, e.g. BMI at enrollment, but all cases were enrolled after a clinical diagnosis of Parkinson's disease was made. How can BMI be 'predictive' when the event has already occurred? Low BMI could be a result of PD (hyperkinetic tremor, swallowing difficulties, nausea from constipation etc), rather than a cause. In the discussion, lines 251-253 it states that being overweight significantly decreased risk of PD, but this is correlation and the study has not proven causation.
- Can the authors show what difference the PRS makes to the final model, e.g. what is the model AUC with and without the PRS included?
- Please expand on the limitations of the study in the discussion section. As well as small numbers, it is a single population, recall bias at interview, Yes/No answers, the AUC of the PRS is only 0.55 suggesting minimal ability to differentiate case from controls. What are the ultimate clinical implications of a model such as this?
- Line 242: What is "SSc"?
- Table 2. Typo "pesticedes" on multiple occasions
- Line 266: "genetic" instead of "genetics"
Author Response
We thank the reviewers for their valuable and constructive comments for the improvement of our manuscript. Based on the reviewers’ comments and suggestions, some extra details were added in the manuscript and some sentences were rephrased and more clearly stated.
Reviewer(s)' Comments to Author:
Reviewer: 1
Point 1: Demographic data is collected at interview, e.g. BMI at enrollment, but all cases were enrolled after a clinical diagnosis of Parkinson's disease was made. How can BMI be 'predictive' when the event has already occurred? Low BMI could be a result of PD (hyperkinetic tremor, swallowing difficulties, nausea from constipation etc), rather than a cause. In the discussion, lines 251-253 it states that being overweight significantly decreased risk of PD, but this is correlation and the study has not proven causation.
Reply:
Indeed BMI was collected at enrolment and its probably difficult to prove causation but rather a finding which could imply a non direct correlation. We clarify this in the manuscript (lines 260-261), just to note also our findings are in concordance with Noyce et al (2017).
Point 2: Can the authors show what difference the PRS makes to the final model, e.g. what is the model AUC with and without the PRS included?
Reply:
The addition of the PRS in the final model made a slight improvement for both the AUC and the GOF [Model without PRS: AUC, 0.78 and GOF, 0.04 and Model with PRS: AUC, 0.79 and GOF, 0.09].It is not a huge improvement but it gives an indication of the additional value that could be achieved using more samples and more SNPs.
Point 3: Please expand on the limitations of the study in the discussion section. As well as small numbers, it is a single population, recall bias at interview, Yes/No answers, the AUC of the PRS is only 0.55 suggesting minimal ability to differentiate case from controls. What are the ultimate clinical implications of a model such as this?
Reply:
Based on this point, limitations and strengths of the study were extended in the last paragraph of the discussion. Indeed, this is a small study with preliminary findings. This will not be the final model that can be used in our population, this needs to be expanded with more individuals and more SNPs and we will also need prospective studies to evaluate its clinical implications. Despite the limitations we believe that this is important to show in our population that the results are in the right direction and the potential when having even a better PRS in our population.
Point 4: Line 242: What is "SSc"?
Reply:
Typo error: It was corrected in this version.
Point 5: Table 2. Typo "pesticedes" on multiple occasions
Reply:
Thank you for the comment. It was corrected in Table 2 and Table S1.
Point 6: Line 266: "genetic" instead of "genetics"
Reply:
It was corrected in this version.

Reviewer 2 Report
In this study, Chairta et al evaluated the predictive performance of a 12-SNPs polygenic risk score (PRS) in combination with environmental/lifestyle factors in a cohort of 235 PD-patients and 464 controls from Cyprus. This 12-SNPs PRS was significantly increased in PD-cases compared to controls and in univariate analyses age, head injury, family history, depression and BMI were significantly associated with PD-status. Step- wise-regression suggested that a model which includes PRS and seven other independent lifestyle/environmental factors was most predictive of PD. They conclude that this PRS could be used in combination with other risk factors for PD risk prediction.
Overall, this is an interesting study and data are well-presented. However, there is some room of improvement.
Major points:
- In determining the PRS, did the authors perform some form of quality check? E.g., did they remove SNPs with comparatively low genotyping rate or minor allele frequency? More important, did they exclude relatedness (1st and 2nd degree relatives), eg. by performing primary component analysis, separately in patients and controls?
- Did the authors check if a PRS containing less than 12 of these SNPs (e.g. 11 or 10) had a better performance than the 12 SNP PRS they used? To perform this shrinkage, they could check if there is linkage disequilibrium between these SNPs or exclude some based on their p value or imputation degree, in the Cypriot cohort and/or in the GWAS cohorts.
- The authors could discuss more in depth the strengths and limitations of their study (last paragraph of the Discussion).
Minor points:
- "we do not claim to have identified something novel just that this method has the potential to be used in the future when we have a larger sample size and more SNPs". This need to be rephrased. Also, they should elaborate on the novelty of the study, that merits publishing.
- "Pesticedes" in Table 2, correct to “pesticides”
- Table 2 could be more succinct, by restructuring it to avoid repeating the same words (eg. Physical Activity, Physical Activity, No Physical Activity)
- Define NR in Table 1.
- Define SSc in Discussion (1st paragraph)
Author Response
We thank the reviewers for their valuable and constructive comments for the improvement of our manuscript. Based on the reviewers’ comments and suggestions, some extra details were added in the manuscript and some sentences were rephrased and more clearly stated.
Reviewer(s)' Comments to Author:
Reviewer: 2
Overall, this is an interesting study and data are well-presented. However, there is some room of improvement.
Major points:
Point 1: In determining the PRS, did the authors perform some form of quality check? E.g., did they remove SNPs with comparatively low genotyping rate or minor allele frequency? More important, did they exclude relatedness (1st and 2nd degree relatives), eg. by performing primary component analysis, separately in patients and controls?
Reply:
Quality control check for all the SNPs and individuals was performed, more details are reported in a previous study carried out by Georgiou et al. (2019). All SNPs and individuals used in the analysis passed the different standard QC filters (call rate and MAF). Deviation from Hardy Weinberg equilibrium was not observed for any of the SNPs (P = 0.15–1). In addition, we were not able to perform formal statistical relatedness tests or ethnicity tests as we only had available information on a small number of SNPs and not genome-wide. All subjects included in the study were self-reported as unrelated. Greek-Cypriot ethnicity of the patients and healthy controls was also self-reported and determined through questionnaires completed by patients/controls. Details of their family trees were also assessed based the questionnaires.
Point 2: Did the authors check if a PRS containing less than 12 of these SNPs (e.g. 11 or 10) had a better performance than the 12 SNP PRS they used? To perform this shrinkage, they could check if there is linkage disequilibrium between these SNPs or exclude some based on their p value or imputation degree, in the Cypriot cohort and/or in the GWAS cohorts.
Reply:
We performed pairwise correlation analysis between the 12 SNPs and found that 11 out of 12 SNPs are unrelated (r between 0.1 and -0.1), thus suggesting that these SNPs are independent and could be combined multiplicatively into the PRS. After your recommendations we evaluated a model using the 11 unrelated SNPs only and the results did not differ substantially compared to the model with 12 SNPs.
Point 3: The authors could discuss more in depth the strengths and limitations of their study (last paragraph of the Discussion).
Reply:
Based on this point, limitations and strengths of the study were extended in the last paragraph of the discussion.
Minor points:
Point 4: "we do not claim to have identified something novel just that this method has the potential to be used in the future when we have a larger sample size and more SNPs". This need to be rephrased. Also, they should elaborate on the novelty of the study, that merits publishing.
Reply:
Thank you for the comment. The previous sentence was rephrased with the following: “Despite that the sample size of our study in this population is relatively small; this study provides important results and will initiate the investigation of combined environmental and genetic factors of PD in the Greek-Cypriot population. Future studies with increased sample size and number of SNPs, could redefine future models and will have the potential to be evaluated for their clinical utility.”
Point 5: "Pesticedes" in Table 2, correct to “pesticides”.
Reply:
Thank you for the comment. It was corrected, but it was removed with the next comment.
Point 6: Table 2 could be more succinct, by restructuring it to avoid repeating the same words (eg. Physical Activity, Physical Activity, No Physical Activity)
Reply:
Table 2 and Table S1 became more succinct. Repeating of the same words was avoided using Yes/No.
Point 7: Define NR in Table 1.
Reply:
NR: Not Reported. It was defined in Table 1.
Point 8: Define SSc in Discussion (1st paragraph)
Reply:
Typo error: It was corrected in this version.
